# The Association between Family Caregiver Burden and Subjective Well-Being and the Moderating Effect of Social Participation among Japanese Adults: A Cross-Sectional Study

**DOI:** 10.3390/healthcare8020087

**Published:** 2020-04-05

**Authors:** Taiji Noguchi, Hiroko Nakagawa-Senda, Yuya Tamai, Takeshi Nishiyama, Miki Watanabe, Mayumi Kamiya, Ryozo Wakabayashi, Akihiro Hosono, Kiyoshi Shibata, Mari Ichikawa, Kanae Ema, Kenji Nagaya, Naoko Okamoto, Shoko Tsujimura, Hitomi Fujita, Fumi Kondo, Tamaki Yamada, Sadao Suzuki

**Affiliations:** 1Department of Public Health, Nagoya City University Graduate School of Medical Sciences, Aichi 467-8601, Japan; noguchi.taiji0415@gmail.com (T.N.);; 2Department of Social Science, Center for Gerontology and Social Science, National Center of Geriatrics and Gerontology, Aichi 474-8511, Japan; 3Department of Nursing, Chukyo Gakuin University, Gifu 509-6192, Japan; 4Atsuta Public Health Center, City of Nagoya, Aichi 456-0031, Japan; 5Department of Health and Nutritional Sciences, Nagoya Keizai University, Aichi 484-8504, Japan; 6Department of Health Nutritional Sciences, Osaka Shoin Women’s University, Osaka 577-8550, Japan; 7Department of Health Sciences, Toyohashi Sozo University, Aichi 440-8511, Japan; 8Department of Rehabilitation, Faculty of Health Sciences, Nihon Fukushi University, Aichi 475-0012, Japan; 9Okazaki City Medical Association, Public Health Center, Aichi 444-0875, Japan

**Keywords:** caregiver burden, subjective well-being, social participation

## Abstract

We examined the association between family caregiver burden and subjective well-being with social participation’s moderating effect among Japanese adults. Data were obtained from a cross-sectional survey by the Japan Multi-Institutional Collaborative Cohort Study in the Okazaki area between 2013 and 2017. Study participants included 5321 adults who visited the Public Health Center for annual health check-ups and answered a questionnaire regarding health status and lifestyle. Subjective well-being was assessed by a single item, out of 10 points, and analyzed with multivariable linear regression analysis models by subjective family caregiver burden (“none”, “mild”, “severe”), stratified by gender. Ultimately, 2857 men and 2223 women were included. Mean participant age (standard deviation) in years was 64.7 (10.4) for men and 61.3 (10.0) for women. Multivariable analysis revealed that, among women, higher caregiver burden was inversely associated with subjective well-being (*p* for trend < 0.001), and the interaction of severe caregiver burden and social participation on subjective well-being was positive and significant (*p* for interaction < 0.05). High family caregiver burden was inversely associated with subjective well-being among Japanese women, but moderated by the caregiver’s social participation, suggesting the importance of community development that enables family caregivers’ social participation to protect their subjective well-being.

## 1. Introduction

Caregiver burden is “the extent to which caregivers perceive that caregiving has had an adverse effect on their emotional, social, financial, physical, and spiritual functioning” [1]. A survey of caregiver burden suggested that 40% of family caregivers have high caregiver burden and 18% have medium caregiver burden, based on time spent providing care [2]. In 2015, an estimated 43.5 million individuals in the United States served as family caregivers to an adult or child, and more than two-thirds of these provided care for an adult older than 50 years [2]. Since an increase in the ageing population inevitably increases the number of individuals living with chronic disease and highlights the inadequacy of formal care services provided to caregivers themselves, a rise in the prevalence and magnitude of caregiver burden can be expected [2].

In Japan, where the ageing rate is the highest in the world, the number of older adults who need nursing care has been increasing rapidly. A public long-term care (LTC) insurance system was initiated in 2000 and, as of 2016, approximately 18% of individuals aged ≥ 65 years are eligible for LTC insurance services [3]. Although the Japanese LTC insurance system was introduced to relieve family caregiver burden [4], the system still depends heavily on informal family care at home, and more than 70% of the LTC services are provided at home [5]. About 43% of family members engage in caregiving for more than two hours each day, and about 22% spend most of the day in caregiving [5]. It is reported that 70% of family caregivers experience troubles and stress [5]. Thus, in Japan, relief from family caregiver burden has not been achieved, and the burden is an important public health issue. 

Family caregiving can be a significant burden on caregivers, and includes heavy assistance with activities of daily living, the decline of social connections and financial deprivation, which can cause psychological distress [6,7]. As a result of these issues, family caregivers often have impaired psychological health [8,9] and impaired subjective well-being [10,11]. Especially because the impairment of happiness is associated with various adverse health conditions [12], the caregivers’ subjective well-being should be protected to prevent the collapse of the care system in Japan, where the majority of care is dependent on family caregiving. In Japan, in particular, there is a strong sense of filial piety toward parents or family members, because of which the burden of family caregiving at home tends to become problematic [13]. However, the Japanese LTC insurance system has limitations in its ability to alleviate family caregiver burden due to the lack of specific nursing care services, decline in family caregiving capacity, and socioeconomic problems faced by family caregivers such as difficulties with managing a job and housework and increasing nursing care costs [14]. Provision of informal resources could be a possible alternative, helping restore family caregivers’ well-being and freeing people receiving care from having to rely too much on public long-term care insurance.

Some studies reported that social participation in the community positively contributes to psychological health [15,16,17]. Acquiring and maintaining social support and social networks could play a role in alleviating caregivers’ psychological distress and in transmitting to them information on making better health and medical choices, which can increase self-esteem and provide a sense of belonging and purpose in life. In particular, family caregivers have been found to often be socially isolated and lonely due to spending time on caregiving [16,18]. Thus, we hypothesized that the social participation of caregivers would moderate the negative association between family caregiver burden and subjective well-being. However, little or nothing has been empirically shown about this possible moderating effect. The purpose of the present study was thus to investigate the moderating effect of social participation on the negative association between family caregiver burden and subjective well-being among Japanese adults.

## 2. Materials and Methods

### 2.1. Study Population

The Japan Multi-Institutional Collaborative Cohort Study (J-MICC Study) was initiated in 2005 with the aim of obtaining fundamental data for the prevention of lifestyle-related diseases, mainly cancer [19,20]. The present cross-sectional study enrolled Japanese adults who participated in the J-MICC Study in the Okazaki area, a suburban area of Japan. Specifically, we considered the data of 5321 individuals (out of the total 7580 asked to participate; response rate 70.2%) who responded to the J-MICC questionnaire (either in person or by mail) when visiting the Okazaki Medical Association Public Health Center in Aichi Prefecture, Japan, for annual health check-ups between 2013 and 2017. We excluded respondents who did not provide information about subjective well-being (88 respondents), presence of care recipients among cohabiting family and resulting caregiver burden (21 respondents), or social participation (11 respondents). Further, we excluded respondents with a Kessler 6 (K6) score of 13 or higher (*n* = 49), recognized as indicating severe depression [21] when assessing psychological distress, and those with missing information of the K6 score (*n* = 72), in order to exclude the respondents with severe mental illness. Ultimately, we analyzed the data of 5080 respondents.

All participants provided written informed consent, and the study protocol was approved by the ethics committee of Nagoya City University Graduate School of Medicine (No. 70-00-0058). The study was conducted in accordance with the guidelines of the Declaration of Helsinki.

### 2.2. Subjective Well-Being

Subjective well-being was assessed using the following single question about subjective happiness, based on a previous study: “Could you place your current sense of happiness on a scale of 100 points?” [22]. We used this score rounded up to 10 to fit the scale of the Japanese National Survey [23], as in previous research [24]. Following earlier research, we used this scale as a measure of subjective well-being [24], because a sense of happiness is considered an essential component of one’s subjective well-being [25,26].

### 2.3. Family Caregiver Burden

To assess subjective caregiver burden, we used two questions: “Do you currently have someone at home who needs nursing care?” (for the presence of care recipients in the cohabiting family) and “How much burden does nursing care cause on yourself, overall?” (for subjective caregiver burden; possible answers: “almost none”, “mild”, “severe”, “very severe”, and “extremely severe”). We classified subjective caregiver burden into three categories: “no caregiver burden” for those having no one at home with nursing care necessities; and “almost none” care burden, “mild caregiver burden” for those with “mild” care burden, and “severe caregiver burden” for those having “severe”, “very severe”, and “extremely severe” care burden.

### 2.4. Social Participation

Social participation was measured by asking, “Do you currently participate in a hobby club, learning, volunteer activities, daily?” Participants could choose their response from “never”, “past”, “sometimes”, and “once a week”. We divided participants into two groups: “social participation” (“sometimes” or “once a week”) and “no participation” (“never” or “past”).

### 2.5. Covariates

Questions on socio-demographic characteristics and health status were included in the analysis as covariates: age, marital status, living arrangement, family relationship, educational attainment, employment status, self-rated health, present illness, and lifestyle. Age was categorized as follows: under 49, 50 to 59, 60 to 69, and 70 years or older. Marital status was categorized as follows: married, divorced/separated, and never married. Living arrangement was categorized as follows: living alone, living with a spouse, living with two or more generations, and other. Family relationship satisfaction was measured by a single question: “Are you satisfied with your family relationship?” The response options were categorized as follows: excellent, good, and not good. Educational attainment was categorized as follows: ≤9, 10 to 12, and ≥13 years. Employment status was categorized as follows: regular employment, irregular employment, not in employment, and other. Self-rated health was measured by a single question: “What is your current health status?” The response options were dichotomized: good and poor. Present illness was assessed by a questionnaire that asked respondents whether they had received cancer, heart disease, and/or stroke diagnoses. Respondents were required to select “yes” or “no” responses. Lifestyle was assessed on smoking and drinking habits. Smoking was dichotomized: never/past and current. Drinking was also dichotomized: never/past and current.

### 2.6. Statistical Analysis

All analyses were conducted according to gender, because we hypothesized that the association between caregiver burden and subjective well-being, and the moderating effect of social participation, would differ by gender [27,28]. First, we presented descriptive statistics. We investigated the differences for each variable based on the severity of the caregiver burden. Second, in order to examine the association between family caregiver burden and subjective well-being, we conducted a multivariable linear regression analysis to obtain unstandardized regression coefficients (βs) and 95% confidence intervals (CIs) for subjective well-being. The analyses were adjusted by age, marital status, living arrangement, family relationship, educational attainment, employment status, self-rated health, present illness, smoking, and drinking (model 1). Next, we added social participation to model 1, yielding model 2. Further, to investigate the moderating effect of social participation on the association between caregiver burden and subjective well-being, we added to model 2 the interaction term of caregiver burden × social participation (code, no participation: 0, and participation: 1), yielding model 3.

To mitigate potential biases caused by missing information, we used the multiple imputation approach, under the missing at random (MAR) assumption (i.e., that the missing data mechanism depends only on the observed variables). We generated 20 imputed data sets using the multiple imputation by chained equations (MICE) procedure and pooled the results using the standard Rubin’s rule [29].

The significance level was set at *p*-value < 0.05. We used R software (Version 3.4.3 for Windows) for all statistical analyses. The multiple imputation approach used the MICE function (mice package).

## 3. Results

A final total of 5080 participants (2857 men and 2223 women) were included. The characteristics of the participants are presented in Table 1. The mean participant age was 64.7 years (standard deviation = 10.4 years) for men and 61.3 years (10.0 years) for women. Among men, for caregiver burden, 2629 (90.0%) participants were classified as “none”, 151 (5.3%) were classified as “mild”, and 77 (2.7%) were classified as “severe”; among women, 1990 (89.5%) were classified as “none”, 149 (5.8%) were classified as “mild”, and 105 (4.7%) were classified as “severe”. Participants with more severe caregiver burden were more likely to be over 60 years, live with others, have poor family relationships, not be in employment, not drink, and have low subjective well-being, among both men and women. While male participants with more caregiver burden tended to engage in social participation, female participants showed the opposite tendency.

The association between family caregiver burden and subjective well-being was seen both among men (Table 2) and women (Table 3). Among men, family caregiver burden was not significantly associated with subjective well-being, adjusted by all covariates (model 1, compared with “none”, “mild”: β = −0.04, SE = −0.1; “severe”: β = −0.19, SE = 0.14; *p* for trend = 0.174). These tendencies did not change after adding social participation (model 2, “mild”: β = −0.05, SE = 0.10; “severe”: β = −0.21, SE = 0.14; *p* for trend = 0.143). Among women, family caregiver burden was inversely associated with subjective well-being (model 1, compared with “none”, “mild”: β = −0.28, SE = 0.11; “severe”: β = −0.46, SE = −0.12; *p* for trend < 0.001), and these tendencies did not change after adding social participation (model 2, “mild”: β = −0.27, SE = 0.11; “severe”: β = −0.45, SE = 0.12; *p* for trend < 0.001). Additionally, to investigate the moderating effect of social participation, we added to model 2 the interaction term of family caregiver burden × social participation, whereby social participation was coded as no participation: 0 or participation: 1 (model 3). Severe caregiver burden × social participation showed a significantly positive interaction on subjective well-being (β = 0.54, SE = 0.24, *p* < 0.05).

Figure 1 shows the mean score of subjective well-being for each level of family caregiver burden stratified by social participation. Among male participants, the higher the burden of caregiving, the lower the subjective well-being score, but there was no difference between those with and those without social participation; while, among women, the higher the burden, the lower the subjective well-being score; this was more pronounced in the group without social participation, while the subjective well-being score was dramatically lower with severe burden.

## 4. Discussion

In the present cross-sectional study, we examined the moderating effect of caregivers’ social participation on the negative association between family caregiver burden and subjective well-being in 5080 Japanese adults. Our study revealed that higher caregiver burden was inversely associated with subjective well-being, but that social participation moderated this association among female participants. To the best of our knowledge, this is the first study to report the moderating effect of social participation on low subjective well-being due to caregiving. Our findings suggest the importance of community development, which can enable family caregivers to participate in community activities and maintain their well-being.

The results of our study showed that higher caregiver burden was associated with lower subjective well-being among women. Previous studies have reported that higher caregiver burden led to impaired psychological health, such as depression [8,9] and low well-being [10,11], due to physical or instrumental provision of aid to care recipients and stress or anxiety [6,7]. Our results supported these previous studies. We believe that in Japan, where public LTC insurance has been introduced to reduce the burden of family caregiving, the family caregiver burden is an important issue that needs more attention. Notably, in Japan, since the culture places importance on filial devotion to parents and family members [13], there is a risk that the burden of family caregiving could become more severe if family members take care of them very often or to a great extent.

In the present study, the association between family caregiver burden and subjective well-being was moderated by caregiver’s social participation among women, which suggests that even if the burden of caregiving is severe, the decline of subjective well-being may be lessened when caregivers have good social participation. There are several potential reasons for these results. First, social participation could help reduce the loneliness of caregivers, which they often experience because time spent caregiving can make it easier to disconnect from society and community [30]. Thus, social isolation could in turn make caregivers less happy. In contrast, through social participation in the community, caregivers can achieve a sense of belonging [17] and alleviate feelings of loneliness. Second, social participation could help caregivers obtain social support [11,31] and build networks, which could help prevent the decline of subjective well-being. Peer support for family care solves care problems, and the stress of care could be buffered by emotional support delivered through social organizations and communities. However, while social participation should help maintain the happiness of family caregivers, its detailed mechanisms need further studying.

Our results showed the gender difference in the association between caregiver burden and subjective well-being. There may be several explanations. First, caregiver burden could differ by gender. Previous studies have shown that levels of caregiver burden and psychological health are significantly higher among women than men [32,33]. Furthermore, male caregivers tended to seek informal and formal support through family care [32]; in this way, male caregivers might have more social resources than female caregivers. We, therefore, speculated that having the opportunity to acquire social support through social participation might help foster or preserve happiness among female caregivers to a greater degree than among males. The present study showed that there are gender differences in the association between caregiver burden and subjective well-being in Japanese adults, which in women at least may be alleviated by social participation.

The present study has several limitations. First, the family caregiver burden was evaluated with only one subjective item. Thus, we may not have effectively assessed the burden of caregiving as a whole, because caregiver burden consists of various aspects, physical, psychological, and financial [1,34]. Further, the perception of caregiving burden may differ according to gender due to the subjectivity of the assessment. In Japanese society, women (especially daughters-in-law) are expected to care for elders at home [13,35]. This cultural context may lead to misclassification of caregiving burden. Second, we may not have been able to adjust for confounders adequately. We did not have information about the kin-relationships between caregivers and care-recipients (for instance, spouse, parent–child, parent–child-in-law), and so could not consider it in the analysis, which may cause potential confounding bias [36]. However, because we tried to reduce this bias by adjusting for satisfaction with family relationships, this effect may be relatively small. Additionally, we did not consider the care recipient’s disease or the level of care required. For instance, family caregivers for individuals with dementia have been reported to have a higher burden and poorer mental health than caregivers for individuals without dementia [37]. Further research considering the detailed background factors for caregiving burden is desirable. Third, our analysis may not have fully controlled for socioeconomic factors. We could not adjust for potential confounders because we had no information about the respondents’ economic situation, which could be associated with caregiving burden and mental health [6,38,39]. However, because we adjusted for employment status and education as socioeconomic factors, this effect too may be relatively small. Fourth, we used a single item on the sense of happiness to evaluate subjective well-being. As subjective well-being is multidimensional, including life evaluation, hedonism, and eudemonic well-being [12], we may not be able to capture subjective well-being holistically. Fifth, our sample was not representative of Japanese adults because the study was conducted in only one suburban area of Japan. Additionally, the participants may have been more concerned about their health than other residents in the area because they were recruited from members of the general population who visited the public health center for annual health check-ups. This may limit the generalizability of our findings. Finally, since the present study was based on a cross-sectional survey, the causal relationships are unclear. Further studies are needed using a longitudinal design to explore the dynamics of the association between family caregiver burden and subjective well-being.

Despite the limitations described above, our findings provide useful suggestions for the protection of well-being among family caregivers. Naturally, it is necessary to aim to reduce the burden on family caregivers by establishing a public LTC system. Further, we believe that policymakers, local government, and care and health professionals need to develop peer support to help family caregivers to provide consultation and support, intervene in community activities to ensure that family caregivers are not isolated, and create communities for this purpose.

## 5. Conclusions

The present study found that higher family caregiver burden was associated with lower subjective well-being among Japanese women. However, social participation by these family caregivers could attenuate the decline of subjective well-being. Our findings suggest the importance of family caregivers’ community participation to receive peer support and stay connected in order to enhance the well-being of family caregivers.

## Figures and Tables

**Figure 1 healthcare-08-00087-f001:**
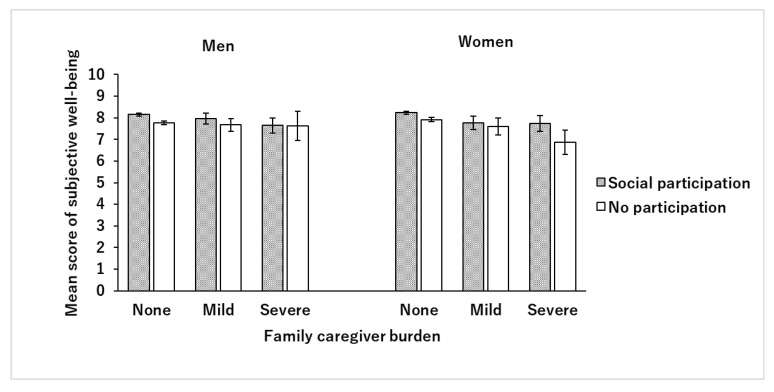
Mean scores of subjective well-being stratified by social participation of family caregivers. The bars in the figure show the mean subjective well-being scores, and the error bars show 95% confidence intervals. Gray bars indicate the scores of those with social participation, and white bars indicate the scores of those without social participation. The figure demonstrates that the higher caregiver burden, the lower the subjective well-being score, in both men and women. In male participants, the trend was almost the same regardless of social participation, while in female participants, in contrast with those with social participation, those with no participation had significantly lower subjective well-being scores when caregiver burden increased.

**Table 1 healthcare-08-00087-t001:** Characteristics of participants.

Variable		Category	Men (*n* = 2857)			Women (*n* = 2223)		
			Family Caregiver Burden			Family Caregiver Burden		
			None (*n* = 2629)	Mild (*n* = 151)	Severe (*n* = 77)	None (*n* = 1990)	Mild (*n* = 128)	Severe (*n* = 105)
Age (years), *n* (%)		≤49	301 (11.4)	7 (4.6)	2 (2.6)	316 (15.9)	14 (10.9)	8 (7.6)
		50 to 59	464 (17.6)	36 (23.8)	8 (10.4)	497 (25.0)	45 (35.2)	25 (23.8)
		60 to 69	808 (30.7)	64 (42.4)	32 (41.6)	702 (35.3)	49 (38.3)	48 (45.7)
		≥70	1056 (40.2)	44 (29.1)	35 (45.5)	475 (23.9)	20 (15.6)	24 (22.9)
Marital status, *n* (%)		Married	2316 (88.1)	130 (86.1)	68 (88.3)	1550 (77.9)	113 (88.3)	83 (79.0)
		Divorced/separated	189 (7.2)	10 (6.6)	4 (5.2)	347 (17.4)	11 (8.6)	13 (12.4)
		Never married	97 (3.7)	11 (7.3)	4 (5.2)	77 (3.9)	3 (2.3)	9 (8.6)
		Missing	27 (1.0)	0 (0.0)	1 (1.3)	16 (0.8)	1 (0.8)	0 (0.0)
Living arrangement, *n* (%)		Living alone	155 (5.9)	1 (0.7)	1 (1.3)	190 (9.5)	1 (0.8)	3 (2.9)
		Living with spouse	1030 (39.2)	27 (17.9)	16 (20.8)	660 (33.2)	19 (14.8)	17 (16.2)
		Living with two or more generations	1411 (53.7)	114 (75.5)	56 (72.7)	1104 (55.5)	106 (82.8)	82 (78.1)
		Other	28 (1.1)	9 (6.0)	4 (5.2)	33 (1.7)	2 (1.6)	3 (2.9)
		Missing	5 (0.2)	0 (0.0)	0 (0.0)	3 (0.2)	0 (0.0)	0 (0.0)
Family relationship, *n* (%)		Excellent	1212 (46.1)	51 (33.8)	25 (32.5)	902 (45.3)	38 (29.7)	26 (24.8)
		Good	1198 (45.6)	84 (55.6)	41 (53.2)	887 (44.6)	74 (57.8)	61 (58.1)
		Not good	183 (7.0)	16 (10.6)	11 (14.3)	176 (8.8)	15 (11.7)	17 (16.2)
		Missing	36 (1.4)	0 (0.0)	0 (0.0)	25 (1.3)	1 (0.8)	1 (1.0)
Educational attainment (years), *n* (%)		≤9	380 (14.5)	16 (10.6)	10 (13.0)	242 (12.2)	15 (11.7)	7 (6.7)
		9 to 12	1095 (41.7)	56 (37.1)	41 (53.2)	844 (42.4)	53 (41.4)	41 (39.0)
		≥13	1153 (43.9)	79 (52.3)	26 (33.8)	904 (45.4)	60 (46.9)	57 (54.3)
		Missing	1 (0.0)	0 (0.0)	0 (0.0)	0 (0.0)	0 (0.0)	0 (0.0)
Employment status, *n* (%)		Regular employment	1137 (43.2)	73 (48.3)	20 (26.0)	541 (27.2)	41 (32.0)	32 (30.5)
		Irregular employment	271 (10.3)	13 (8.6)	4 (5.2)	406 (20.4)	37 (28.9)	18 (17.1)
		Not in employment	1052 (40.0)	53 (35.1)	50 (64.9)	949 (47.7)	43 (33.6)	53 (50.5)
		Other	161 (6.1)	12 (7.9)	3 (3.9)	90 (4.5)	7 (5.5)	1 (1.0)
		Missing	8 (0.3)	0 (0.0)	0 (0.0)	4 (0.2)	0 (0.0)	1 (1.0)
Present illness, *n* (%)	Cancer	No	2350 (89.4)	133 (88.1)	71 (92.2)	1772 (89.0)	119 (93.0)	90 (85.7)
		Yes	143 (5.4)	11 (7.3)	5 (6.5)	59 (3.0)	2 (1.6)	3 (2.9)
		Missing	136 (5.2)	7 (4.6)	1 (1.3)	159 (8.0)	7 (5.5)	12 (11.4)
	Heart disease	No	2456 (93.4)	135 (89.4)	73 (94.8)	1910 (96.0)	126 (98.4)	99 (94.3)
		Yes	96 (3.7)	10 (6.6)	3 (3.9)	23 (1.2)	0 (0.0)	1 (1.0)
		Missing	77 (2.9)	6 (4.0)	1 (1.3)	57 (2.9)	2 (1.6)	5 (4.8)
	Stroke	No	2469 (93.9)	140 (92.7)	73 (94.8)	1870 (94.0)	125 (97.7)	96 (91.4)
		Yes	36 (1.4)	4 (2.6)	1 (1.3)	17 (0.9)	1 (0.8)	0 (0.0)
		Missing	124 (4.7)	7 (4.6)	3 (3.9)	103 (5.2)	2 (1.6)	9 (8.6)
Self-rated health, *n* (%)		Poor	474 (18.0)	24 (15.9)	15 (19.5)	424 (21.3)	29 (22.7)	34 (32.4)
		Good	2149 (81.7)	126 (83.4)	62 (80.5)	1562 (78.5)	98 (76.6)	71 (67.6)
		Missing	6 (0.2)	1 (0.7)	0 (0.0)	4 (0.2)	1 (0.8)	0 (0.0)
Smoking, *n* (%)		Never/past	2184 (83.1)	119 (78.8)	66 (85.7)	1874 (94.2)	122 (95.3)	102 (97.1)
		Current	442 (16.8)	31 (20.5)	11 (14.3)	79 (4.0)	5 (3.9)	0 (0.0)
		Missing	3 (0.1)	1 (0.7)	0 (0.0)	37 (1.9)	1 (0.8)	3 (2.9)
Drinking, *n* (%)		Never/past	807 (30.7)	50 (33.1)	33 (42.9)	1300 (65.3)	83 (64.8)	71 (67.6)
		Current	1817 (69.1)	100 (66.2)	44 (57.1)	672 (33.8)	44 (34.4)	31 (29.5)
		Missing	5 (0.2)	1 (0.7)	0 (0.0)	18 (0.9)	1 (0.8)	3 (2.9)
Social participation, *n* (%)		No participation	1211 (46.1)	71 (47.0)	29 (37.7)	715 (35.9)	54 (42.2)	45 (42.9)
		Participation	1418 (53.9)	80 (53.0)	48 (62.3)	1275 (64.1)	74 (57.8)	60 (57.1)
Subjective well-being, mean (SD)			7.98 (1.35)	7.83 (1.19)	7.64 (1.43)	8.12 (1.32)	7.69 (1.39)	7.37 (1.68)

SD, standard deviation.

**Table 2 healthcare-08-00087-t002:** The association between family caregiver burden and subjective well-being, multivariable linear regression analysis (men).

Variable	Category	Model 1			Model 2			Model 3		
		β		SE	β		SE	β		SE
Family caregiver burden	None	Reference			Reference			Reference		
	Mild	−0.04		0.10	−0.05		0.10	0.04		0.15
	Severe	−0.19		0.14	−0.21		0.14	−0.07		0.23
Social participation	No participation				Reference			Reference		
	Participation				0.25	***	0.05	0.27	***	0.05
Family caregiver burden × social participation	Mild × participation							−0.17		0.20
	Severe × participation							−0.22		0.29
Age (years)	≤49	Reference			Reference			Reference		
	50 to 59	0.16	†	0.09	0.17	†	0.09	0.17	†	0.09
	60 to 69	0.32	***	0.09	0.30	**	0.09	0.29	**	0.09
	≥70	0.30	**	0.10	0.24	**	0.10	0.24	**	0.10
Marital status	Married	Reference			Reference			Reference		
	Divorced/separated	−0.20	*	0.10	−0.20	*	0.10	−0.21	*	0.10
	Never married	−0.66	***	0.13	−0.63	***	0.13	−0.64	***	0.13
Living arrangement	Living alone	Reference			Reference			Reference		
	Living with spouse	0.36	**	0.13	0.35	**	0.13	0.34	**	0.13
	Living with two or more generations	0.38	**	0.12	0.36	**	0.12	0.36	**	0.12
	Other	0.06		0.22	0.08		0.22	0.08		0.22
Family relationship	Excellent	Reference			Reference			Reference		
	Good	−0.70	***	0.05	−0.69	***	0.05	−0.69	***	0.05
	Not good	−1.64	***	0.09	−1.64	***	0.09	−1.65	***	0.09
Educational attainment (years)	≤9	Reference			Reference			Reference		
	10 to 12	-0.04		0.07	-0.05		0.07	-0.05		0.07
	≥13	−0.05		0.07	−0.06		0.07	−0.06		0.07
Employment status	Regular employment	Reference			Reference			Reference		
	Irregular employment	0.00		0.09	0.00		0.09	0.00		0.09
	Not in employment	−0.02		0.07	−0.06		0.07	−0.06		0.07
	Other	0.08		0.11	0.07		0.10	0.07		0.10
Cancer	No	Reference			Reference			Reference		
	Yes	−0.01		0.10	0.00		0.10	0.00		0.10
Heart disease	No	Reference			Reference			Reference		
	Yes	0.10		0.12	0.12		0.12	0.12		0.12
Stroke	No	Reference			Reference			Reference		
	Yes	0.05		0.18	0.06		0.18	0.06		0.18
Self-rated health	Poor	Reference			Reference			Reference		
	Good	0.59	***	0.06	0.57	***	0.06	0.57	***	0.06
Smoking	Never/past	Reference			Reference			Reference		
	Current	0.06		0.06	0.07		0.06	0.07		0.06
Drinking	Never/past	Reference			Reference			Reference		
	Current	-0.02		0.05	-0.04		0.05	-0.03		0.05

*, *p* < 0.5; **, *p* < 0.01; ***, *p* < 0.001; †, *p* < 0.1. β, unstandardized regression coefficient; SE, standard error.

**Table 3 healthcare-08-00087-t003:** The association between family caregiver burden and subjective well-being, multivariable linear regression analysis (women)

Variable	Category	Model 1			Model 2			Model 3		
		β		SE	β		SE	β		SE
Family caregiver burden	None	Reference			Reference			Reference		
	Mild	−0.28	*	0.11	−0.27	*	0.11	−0.18		0.17
	Severe	−0.46	***	0.12	−0.45	***	0.12	−0.76	***	0.18
Social participation	No participation				Reference			Reference		
	Participation				0.19	***	0.06	0.17	**	0.06
Family caregiver burden × social participation	Mild × participation							−0.15		0.22
	Severe × participation							0.54	*	0.24
Age (years)	≤49	Reference			Reference			Reference		
	50 to 59	−0.09		0.08	−0.10		0.08	−0.10		0.08
	60 to 69	0.09		0.09	0.06		0.09	0.06		0.09
	≥70	0.07		0.10	0.02		0.11	0.03		0.11
Marital status	Married	Reference			Reference			Reference		
	Divorced/separated	−0.17	*	0.09	−0.17	*	0.09	−0.18	*	0.09
	Never married	−0.41	**	0.14	−0.40	**	0.14	−0.39	**	0.14
Living arrangement	Living alone	Reference			Reference			Reference		
	Living with spouse	0.11		0.12	0.12		0.12	0.11		0.12
	Living with two or more generations	−0.02		0.11	−0.01		0.11	−0.01		0.11
	Other	0.18		0.22	0.20		0.22	0.21		0.22
Family relationship	Excellent	Reference			Reference			Reference		
	Good	−0.59	***	0.05	−0.59	***	0.05	−0.59	***	0.05
	Not good	−1.77	***	0.09	−1.77	***	0.09	−1.77	***	0.09
Educational attainment (years)	≤9	Reference			Reference			Reference		
	10 to 12	−0.03		0.09	−0.06		0.09	−0.06		0.09
	≥13	0.04		0.09	0.00		0.09	0.00		0.09
Employment status	Regular employment	Reference			Reference			Reference		
	Irregular employment	−0.02		0.07	−0.03		0.07	−0.02		0.07
	Not in employment	0.07		0.07	0.04		0.07	0.04		0.07
	Other	0.00		0.13	-0.02		0.13	−0.02		0.13
Cancer	No	Reference			Reference			Reference		
	Yes	−0.14		0.16	−0.11		0.16	−0.11		0.16
Heart disease	No	Reference			Reference			Reference		
	Yes	0.13		0.25	0.13		0.25	0.12		0.25
Stroke	No	Reference			Reference			Reference		
	Yes	−0.23		0.29	−0.20		0.29	−0.21		0.29
Self-rated health	Poor	Reference			Reference			Reference		
	Good	0.56	***	0.06	0.55	***	0.06	0.55	***	0.06
Smoking	Never/past	Reference			Reference			Reference		
	Current	−0.09		0.14	−0.07		0.14	−0.08		0.14
Drinking	Never/past	Reference			Reference			Reference		
	Current	0.00		0.05	−0.01		0.05	−0.01		0.05

*, *p* < 0.5; **, *p* < 0.01; ***, *p* < 0.001; *p* < 0.1. β, unstandardized regression coefficient; SE, standard error.

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
