# Peer review of "The Association between Family Caregiver Burden and Subjective Well-Being and the Moderating Effect of Social Participation among Japanese Adults: A Cross-Sectional Study"

_healthcare, 2020, doi:10.3390/healthcare8020087_

Round 1

Reviewer 1 Report

The association between family caregiver burden and subjective well-being and the moderating effect of social participation among Japanese adults: A cross-sectional study

Overall comments

This study used the data from Japan Multi-Institutional Collaborative Cohort Study to investigate the linkage between family caregiver burden and subjective well-being and possible moderating role by social participation. I think the topic is interesting and this paper is well-written and quite easy to follow.

However, I still have some minor comments detailed below.

1. Regarding introduction, authors mentioned the number of individuals in the U.S. served as family caregivers to an adult or a child. Well, given that the research area of this present study is in the Okazaki area of Japan, I am curious about whether authors could provide some related information. That would strengthen the significance of this study.

In addition, authors highlighted that Japan’s ageing rate is the highest worldwide. It is better to add more substance on the prevalence of old adults aged 60 or 65 years old and over and the projected prevalence in future based on the United Nations. I think it would also make the background more solid.

2. For materials and methods, authors only focus on the Okazaki area, which makes the estimated results impossible to generalize to the whole Japan. The external validity is a major flaw in this study.

My concern also goes to the measure of subjective well-being (SWB). As we know, SWB is multidimensional. Authors adopted happiness as one proxy of SWB, which is, in essence, is a measure of hedonic well-being capturing the emotional quality of everyday experience. I just wonder whether other measures of SWB are available, for instance, life satisfaction, depression or others. If this is the case, a combination of multiple measures should be used. This also raise another limitation in this study: just use one-item measure of SWB. I think authors have to mention this in the limitations.

3. As regards discussion, authors have to confess that due to the use of cross-sectional design, it is impossible to identify the causal relationship between family caregiver burden and SWB. In addition, it is also worth highlighting that more studies using longitudinal data to explore the dynamics of the linkage between family caregiver burden and SWB is needed.

Author Response

Reviewer 1

Overall comments:

This study used the data from Japan Multi-Institutional Collaborative Cohort Study to investigate the linkage between family caregiver burden and subjective well-being and possible moderating role by social participation. I think the topic is interesting and this paper is well-written and quite easy to follow.

Response:

Thank you for the thoughtful and constructive feedback on our manuscript. We hope that our revisions adequately address all comments.

Comment #1-1:

Regarding introduction, authors mentioned the number of individuals in the U.S. served as family caregivers to an adult or a child. Well, given that the research area of this present study is in the Okazaki area of Japan, I am curious about whether authors could provide some related information. That would strengthen the significance of this study.

Response #1-1:

Thank you for the useful advice. We have added information on the current state of care in Japan, especially for older adults (lines, 57-64): “In Japan, where the ageing rate is the highest in the world, the number of older adults who need nursing care has been increasing rapidly. A public long-term care (LTC) insurance system was initiated in 2000 and as of 2016, approximately 18% of the individuals aged ≥ 65 years are eligible for LTC insurance services [3]. Although the Japanese LTC insurance system were introduced to relieve family caregiver burden [4], the system still depends heavily on informal family care at home, and more than 70% of the LTC services are provided at home [5]. About 43% family members engage in caregiving for more than two hours each day, and about 22% spend most of the day in caregiving [5]. It is reported that 70% of family caregivers experience troubles and stress [5].”

Comment #1-2:

In addition, authors highlighted that Japan’s ageing rate is the highest worldwide. It is better to add more substance on the prevalence of old adults aged 60 or 65 years old and over and the projected prevalence in future based on the United Nations. I think it would also make the background more solid.

Response #1-2:

Thank you for your advice. In connection with the previous comment (Comment #1-1), We have added information on the situation of Japanese older people aged 65 and over (lines, 57-60): “In Japan, where the ageing rate is the highest in the world, the number of older adults who need nursing care has been increasing rapidly. A public long-term care (LTC) insurance system was initiated in 2000 and as of 2016, approximately 18% of the individuals aged ≥ 65 years are eligible for LTC insurance services [3].”

Comment #2-1:

For materials and methods, authors only focus on the Okazaki area, which makes the estimated results impossible to generalize to the whole Japan. The external validity is a major flaw in this study.

Response #2-1:

Thank you for your advice. We have clarified that our study was conducted in the Okazaki area, which is a suburban city of Japan, and added the impact on generalizability of our results in Materials and Methods and the study limitations. (Materials and Methods: lines, 96-97): “The present cross-sectional study enrolled Japanese adults who participated in the J-MICC Study in the Okazaki area, a suburban area of Japan.” and (Discussion: lines, 286-287): “Fifth, our sample was not representative of Japanese adults because the study was conducted in only one suburban area of Japan.”

Comment #2-2:

My concern also goes to the measure of subjective well-being (SWB). As we know, SWB is multidimensional. Authors adopted happiness as one proxy of SWB, which is, in essence, is a measure of hedonic well-being capturing the emotional quality of everyday experience. I just wonder whether other measures of SWB are available, for instance, life satisfaction, depression or others. If this is the case, a combination of multiple measures should be used. This also raise another limitation in this study: just use one-item measure of SWB. I think authors have to mention this in the limitations.

Response #2-2:

Thank you for the important suggestions. Subjective well-being is multidimensionally structured and evaluated by approaches such as life evaluation, hedonic well-being, and eudemonic well-being (Steptoe A. et al., Lancet. 2015). We used a single item on the sense of happiness as a proxy indicator of subjective well-being. Happiness is one aspect of subjective well-being, but we may not be able to capture the whole subjective well-being. Thus, we have added this point to the study limitations (lines, 283-286): “Fourth, we used a single item on the sense of happiness to evaluate subjective well-being. As subjective well-being is multidimensional, including life evaluation, hedonism, and eudemonic well-being [12], we may not be able to capture subjective well-being holistically.”

Comment #3:

As regards discussion, authors have to confess that due to the use of cross-sectional design, it is impossible to identify the causal relationship between family caregiver burden and SWB. In addition, it is also worth highlighting that more studies using longitudinal data to explore the dynamics of the linkage between family caregiver burden and SWB is needed.

Response #3:

Thank you for your comments. We have revised the study limitations on cross-sectional study design (lines, 291-293): “Further studies are needed using a longitudinal design to explore the dynamics of the association between family caregiver burden and subjective well-being.”

Reviewer 2 Report

I found the article and study of interest as it falls within my research field and I would like to congratulate the authors on a very good study.

I do feel that the study and in particular the Introduction should be language edited to improve clarity of statements. (I am attaching the document with highlighted comments.

The methodology was excellently executed and well-explained.

I would like to make one small suggestion and that is that the Conclusion be expanded to include suggestions of types of community participation, as it can mean further caregiver work OR leisure time for the participants.

More specific issues:

1)  Line 35, 141, 143 - Please refer to "gender" and not "sex" as it is the more common use.

2)  Line 59, 60 - Please restructure sentence/needs editing. It is confusing.

3)  Line 62, 63. 64 - Needs editing

4)  Line 65, 66 - "...the caregivers subjective well-being should be protected."  Please explain or expand this sentence?

5)  Line 81 - Please remove "to the best of our knowledge"  The rest of the sentence is fine and will suffice.

6)  Line 99 - Replace "In the end..." with "Ultimately"

7)  Line 169 - "those with and without"  should be changed to "those with and those without..."

8)  Line 201 to 206:  Figure 1 should have a title and the paragraph be moved to above he table and be part of the paragraph before.

9)  Line 214 - Please elaborate on "strong communities" and what it entails as opposed to other types of communities.  This could be a limitation of the study too, as the authors could comment on the specific community in which the study was done.

10) Line 240, 241 - Repetitive statement.

11) Line 282, 293 - Please elaborate on community participation as mentioned above.

Author Response

Reviewer 2

Comment #1:

I found the article and study of interest as it falls within my research field and I would like to congratulate the authors on a very good study.

I do feel that the study and in particular the Introduction should be language edited to improve clarity of statements. (I am attaching the document with highlighted comments.

The methodology was excellently executed and well-explained.

Response #1:

Thank you for the useful feedback on our manuscript. We revised our manuscript according to your comment. We hope that our revisions fully address your concerns.

Comment #2:

I would like to make one small suggestion and that is that the Conclusion be expanded to include suggestions of types of community participation, as it can mean further caregiver work OR leisure time for the participants.

Response #2:

Thank you for providing important insights. For instance, receiving peer support can solve problems in family caregiving, and maintaining ties with the community can expect to moderate loneliness. Therefore, we believe that there is a need for types of community participation that can provide caregivers these benefits. We have added a proposal on the types of social participation to the “Conclusions” (lines, 303-305): “Our findings suggest the importance of family caregivers’ community participation to receive peer support and stay connected in order to enhance the well-being of family caregivers.”

Comment #3:

More specific issues:

1) Line 35, 141, 143 - Please refer to "gender" and not "sex" as it is the more common use.

Response #3-1):

We corrected the relevant parts to “gender” (lines 36, 149, and 151).

2) Line 59, 60 - Please restructure sentence/needs editing. It is confusing.

Response #3-2):

We revised this sentence (lines, 61-62): “…the system still depends heavily on informal family care at home, and more than 70% of the LTC services are provided at home [5].”

3) Line 62, 63. 64 - Needs editing

Response #3-3):

We revised this sentence (lines, 67-69): “Family caregiving can be a significant burden on caregivers, including heavy assistance with activities of daily living, the decline of social connections and financial deprivation can cause psychological distress [6,7].”

4) Line 65, 66 - "...the caregivers subjective well-being should be protected. "Please explain or expand this sentence?

Response #3-4):

As you pointed out, we added explanations of this part (lines, 71-73): “…the caregivers’ subjective well-being should be protected to prevent the collapse of the care system in Japan, where the majority of care is dependent on family caregiving.”

5) Line 81 - Please remove "to the best of our knowledge "The rest of the sentence is fine and will suffice.

Response #3-5):

Thank you for your advice. We deleted this part (line, 88).

6) Line 99 - Replace "In the end..." with "Ultimately"

Response #3-6):

We corrected this part (line, 106).

7) Line 169 - "those with and without "should be changed to "those with and those without..."

Response #3-7):

We corrected this part (lines, 210).

8) Line 201 to 206: Figure 1 should have a title and the paragraph be moved to above he table and be part of the paragraph before.

Response 3-8):

We clarified the title of the figure (line, 215): “Figure 1. Mean scores of subjective well-being stratified by social participation of family caregivers.”

On the other hand, because we added the text in Introduction, the paragraph associated with this figure was placed below the tables (lines, 208-213).

9) Line 214 - Please elaborate on "strong communities" and what it entails as opposed to other types of communities. This could be a limitation of the study too, as the authors could comment on the specific community in which the study was done.

Response #3-9):

Thank you for your comment. We thought the term “strong communities” could be a bit misleading, so we deleted “strong.” This is because our suggestion is that need for communities to allow family caregivers to participate in community organizations, not whether they are “strong.” (line, 231): “Our findings suggest the importance of community development, which can enable family caregivers to participate in community activities and maintain their well-being.”

10) Line 240, 241 - Repetitive statement.

Response #3-10):

We revised these sentences (lines, 255-256): “Our results showed the gender difference in the association between caregiver burden and subjective well-being. There may be several explanations.”

11) Line 282, 293 - Please elaborate on community participation as mentioned above.

Response #3-11):

We have added a proposal on the types of social participation to the “Conclusions” (lines, 303-305): “Our findings suggest the importance of family caregivers’ community participation to receive peer support and stay connected in order to enhance the well-being of family caregivers.”

This manuscript is a resubmission of an earlier submission. The following is a list of the peer review reports and author responses from that submission.

Round 1

Reviewer 1 Report

Scientific evidence to provide effective suggestions to improve caregivers’ daily life and their subjective well-being is essential, regarding their relevant social role.

Several topics of the manuscript would profit with further exploration, thorough revision.

Some reflections reading this manuscript, I hope you will find helpful:

Additional keyword seems relevant: Japan.

The introduction does not clearly describe the context of this research for supplementary comprehension (for instance: How many caregivers in Japan? Is there any legislative support? Until what extent public policies address their demands? Who does profit with caregiver task? What is the dimension of the gender gap level? And so on).

In materials and methods, the reader would benefit with more detailed information on the study population (e.g. Who are people that participated in J-MICC Study? Are or aren’t informal caregivers? Etc.). Please characterize briefly J-MICC population.

Table 2 is missing. Perhaps it is the second “Table 1”, in line 182, page 6.

Line 243, again, do you have numbers on males and females caregivers’ gap?

Line 266 until 269, please explain suggestion. How could policymakers and professionals enable caregivers’ opportunities for social participation? Could you give specific and real recommendations?

Discussion and conclusions consist of a weakly reflection and analysis. A deeper exploration is required. Perhaps a mix methods research design could have improved debate. 

Reviewer 2 Report

This paper addresses an important issue as with population ageing and longevity, the period of care-giving for adult family members (caregivers) has increased considerably. 

Abstract:

Does not address the significance of the issue in the global/ high income societies nor specifically in the country the study was conduced. It largely comprises of Findings {results) and has no section on implications. 

Introduction:

Some of the relevant literature is outlined. The reader does not get a sense of the burden of care-giving and of societal expectations in Japan. 

In many countries, there is evidence that although a large proportion of primary carers are women, but this dynamic is changing both within the care-recipient couple as well as for daughters and sons. It would have been helpful to discuss the care-giving issues in a more holistic manner for Japanese society and how the concept of filial piety may be changing.  

Methods:

Whilst the authors have mentioned other studies that show association between poor care-giver health and heavy burden of care-giving, both physical and mental; it is not clear as to why the authors chose a single item of a generic nature to assess caregiver health, with no further questions on physical, emotional and mental health.   

The primary outcome variable is based on one single question: "Place your current sense of happiness in [sic] a scale of 100 points".

No scientific rationale is provided how "happiness" equates with "well-being". I do not think in any language the two terms are synonymous. Well-being is a much broader concept and is well-researched topic in social sciences as well as population-level health studies. Please refer to the relevant literature on happiness and well-being. 

Moreover, a single question on happiness with a 100-point score divided into quartile ranges with no other measure to verify, is problematic and scientifically unsound.

Furthermor: the classification into low and high is then done of the basis of "low" being first quartile scores; and high being second, third and fourth quartile scores.  

There are large number of confounders that have not been taken into account. For example, problems with family relationship. How is family defined? The stressors for unhappiness go far beyond marital relationship.

Employment: In today's world of myriad workplace fiscal issues and casualisation of the workforce in many sectors; a binary division between employed and unemployed is highly problematic. People could be under-employed, which is a stressor or currently unemployed as they transition between casual jobs. More importantly, it seems the employment category also include people who are retired as your study sample includes age range <49 years to 80 years and older. 

Care-giving: There is no sound explanation provided for the categorisation: mild and severe. Surely there are carers who have moderate level of carer responsibility.  

No information is provided on Ethics approval for this study. 

Results:

Neither in the text nor in the table is there appropriate information on the sample such as age profile and other demographic variables. Looking at categories of care-giving, the number of respondents in the analysis sample for mild and severe care-giving categories. This in turn makes the multivariable analysis quite unsound. 

Discussion

Many statements are not valid as the results are not scientifically sound. 

Reviewer 3 Report

First of all, I would like to congratulate the authors on the chosen topic. I think it is of vital social and research importance.
Article strengths:

The introduction is adequate. The number of participants in the study. The statistical analyses used.

The weak points:
1. The instruments used are inadequate and lack evidence of reliability and validity.
2. The previous weak point greatly weakens the conclusions of the study.